# The Effect of Size and Strain on Micro Stripe Magnetic Domain Structure of CoFeB Thin Films

Hongyang Li [1,2], Yali Xie [2,*], Huali Yang [2,*], Haixu Hu [2], Mengchao Li [2] and Run-Wei Li [2]

1   School of Material Science and Chemical Engineering, Ningbo University, Ningbo 315211, China
2   CAS Key Laboratory of Magnetic Materials and Devices, Ningbo Institute of Materials Technology
    and Engineering, Chinese Academy of Sciences, Ningbo 315201, China
*   Correspondence: xieyl@nimte.ac.cn (Y.X.); yanghl@nimte.ac.cn (H.Y.)

**Abstract:** The prerequisite for flexible magnetic electronic devices is the knowledge of the preparation technology of flexible magnetic films and the evolution of the film properties under strain. In this work, CoFeB amorphous ferromagnetic films with stripe domains were prepared on flexible polyimide (PI) substrates by oblique sputtering. The results show that oblique sputtering induces the formation of columnar crystal structure in CoFeB films, which increases the perpendicular magnetic anisotropy of the films, thus leading to the appearance of stripe magnetic domain structures. On this basis, the CoFeB films with stripe domains were processed on a microscopic scale to investigate the size effect and strain regulation on the microscopic domain structure of the magnetic films. The characterization of the magnetic domain structure shows that the stripe domain contrast is reduced by the striped structure prepared by lithography. The triangular, circular and ring patterns deflect the alignment of the stripe domain to different degrees. The experimental results show that the deflection of the stripe domains is caused by the anisotropy of the shapes produced by the different patterns and that the size of the microstructure needs to be close to the period of the stripe domains for the size effect to be significant. In addition, the strain-induced magnetoelastic anisotropy effectively rotates the orientation of the stripe domains, and the variation in domain contrast demonstrates that tensile/compressive strains vary the magnitude of the out-of-plane stray field of the film. Our results provide some insight into the modulation of the physical properties of flexible magnetic films.

**Keywords:** flexible magnetic film; stripe magnetic domain; oblique sputtering; magnetoelastic coupling effect

## 1. Introduction

Flexible electronics is an emerging electronic technology that fabricates electronic devices of organic/inorganic materials on flexible/ductile polymers or ultrathin metal substrates. Unlike traditional silicon-based semiconductors with rigid substrates and electronic materials, flexible electronic materials [1–7] have higher flexibility and wider application scope due to their unique flexibility [8] and ductility [9,10]. With the rise of technologies such as healthcare, smart clothing and human-computer interaction, the field of flexible electronics has received wide attention and made long-term progress [11–15]. Among them, magnetic sensing and memory devices are one of the important branches of electronic devices [16–18], and mastering the preparation process of flexible magnetic films [19–24] is an important prerequisite for the development of flexible magnetic sensing and memory devices.

When magnetic films are grown on flexible substrates, the films are subjected to different magnitudes of tensile or compressive strain from the substrates [25,26] when the substrates are in different bending and stretching states, which can cause changes in the magnetic anisotropy [27,28], magnetic domains [29,30], electrical transport [31,32], ferromagnetic resonance frequencies [33,34] and other properties of the magnetic films.

Due to the magnetoelastic coupling effect of the material itself, the magnetic anisotropy of the film changes with the strain state. For instance, Dai et al. [35] confirmed that the easy magnetization axis associated with the magnetic anisotropy tends to be along the tensile strain direction by fixing a flexible substrate on a bending mold and applying strain to the FeGa film on the substrate. For a more complicated case, Berkem et al. [36] prepared Co/Cu/Ni pseudo-spin valves on flexible polyimide substrates. Since Co and Ni have positive and negative magnetostriction coefficient, respectively, the application of uniaxial strain rotates the magnetization of the two ferromagnetic layers in the opposite direction, which leads to an increase in the film magnetoresistance value. Furthermore, Chen et al. [31] investigated the electrical transport properties of the Co film on the flexible polyester substrate, and found that the anisotropic magnetoresistance (AMR) response of the film in the large strain state was significantly larger when an external field was applied along the easy axis direction.

The effect of strain on the macroscopic properties of thin films is further reflected in the microscopic physical properties of the films. For example, Peng et al. [37] investigated the effect of tensile strain on the magnetic domains of FeCoSiB amorphous films. With the increase of tensile strain, the magnetic domain structure was able to change from an originally disordered irregular structure to a striped structure parallel to the strain. Dai et al. [38] found that the striped domains of FeGa films with large magnetostriction coefficients tend to align parallel to the direction of tensile strain in the absence of an applied magnetic field, while the opposite is true under compressive strain. However, when flexible magnetic films are further miniaturized to fit the trend of miniaturization of electronic devices, the size effect of miniaturization on the regulation of magnetic domains and magnetic anisotropy [39–41] has not yet been thoroughly studied. Previous work has reported on the use of electron beams to etch nanogrooves, where due to the difference in thermal expansion coefficients between Ni and the substrate during deposition, the nanogrooves generate tensile stresses which in turn affect the alignment of the magnetic domains [30]. In this work, CoFeB amorphous ferromagnetic films, which can form interfacial perpendicular magnetic anisotropy effects [42,43] and exhibit typical stripe domains by oblique sputtering, were selected as the object of study. The obtained flexible CoFeB does not rely on rigid Si substrates for electron beam exposure, so that strain can be tuned by varying the experimental conditions. Multiple types of patterns were prepared in a single run, while the regulation of the magnetic domains and magnetic anisotropy by the shape anisotropy generated by the different patterns was investigated to provide a basis for the development of miniaturized flexible magnetic devices.

## 2. Materials and Methods

The CoFeB amorphous ferromagnetic films were grown on flexible polyimide (PI) substrates by DC magnetron sputtering at room temperature (RT). Polyimide has good mechanical strength and a high Young's modulus. Metal films deposited on polyimide are less prone to peeling or fracture and have essentially the same properties as those obtained on rigid substrates. In contrast, metal films deposited on polydimethylsiloxane, another polymer, are prone to wrinkling. Polyimide as a substrate will also facilitate our micronano-processing at a later stage, for example, when the sample will experience higher temperatures during ion beam etching. $Co_{40}Fe_{40}B_{20}$ target with 99.95% purity and 50.8 mm diameter was used, which is produced by Hefei kejing materials technology co., LTD (Hefei, China). The base pressure of the sputtering chamber was below $2 \times 10^{-5}$ Pa. The sputtering gas was high-purity argon (99.99% purity), and the argon gas flow was maintained at 20 sccm, and the sputtering pressure was 0.15 Pa. The sputtering power was 160 W. The distance between the sputtering target position and the sample stage is approximately 18 cm. No magnetic field is applied during the deposition process and the rotation of the sample stage is switched off. The film growth rate was first calibrated by growing the film for a constant time (10 min), the sputtering rate was measured to be approximately 5 nm/min. The thickness of the film is varied by controlling the growth

time. A thin Ta layer (∼4 nm) was grown at the top of all samples as a protective layer to prevent sample oxidation.

The hysteresis loops were measured using a vibrating sample magnetometer (VSM, Lakeshore 7410, Columbus, OH, USA). The surface morphology and sputtering rate of the samples were characterized by atomic force microscopy (AFM, Dimension Icon, Bruker, Billerica, MA, USA), and the magnetic domain structure of the samples was characterized by magnetic force microscopy (MFM, Dimension Icon, Bruker, Billerica, MA, USA). The cross-sectional structure of the samples was characterized by scanning electron microscopy (SEM, Zeiss Sigma 300, Oberkochen, Germany).

## 3. Results

To investigate the effect of oblique sputtering on the magnetic anisotropy of CoFeB amorphous magnetic films, CoFeB films with a thickness of 450 nm were grown by adding wedge-shaped molds with different oblique angles to the magnetron sputtering sample tray and changing the angle between the PI substrate normal direction and the incident atomic beam direction (Figure 1a), using an oblique angle $\theta$ of 60°. Typically, homogeneous CoFeB amorphous films grown without an applied magnetic field, stress and without the use of oblique sputtering are magnetically isotropic. Amorphous films do not have to take into account the effects of magneto-crystalline anisotropy, which is common in crystalline films. In contrast, according to a previous report [44], the in-plane uniaxial magnetic anisotropy of CoFeB films increases progressively with increasing oblique sputtering angle. In our work, the hysteresis loops at different magnetic field directions are shown in Figure 1c. The oblique sputtering induces uniaxial magnetic anisotropy in the film plane. The easy axis (EA) direction is parallel to the projection direction of the incident beam on the film plane, while the hard axis (HA) direction is perpendicular to the projection direction of the incident beam on the film plane.

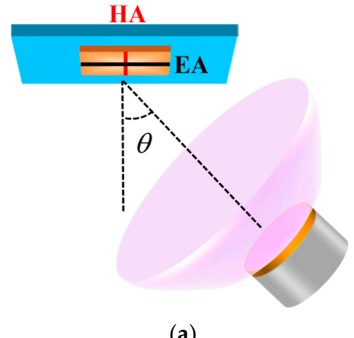

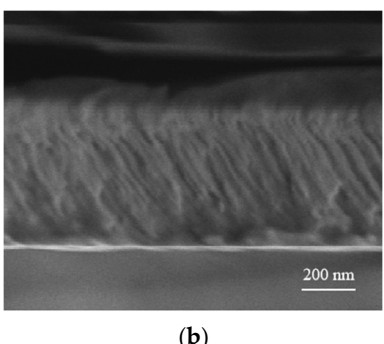

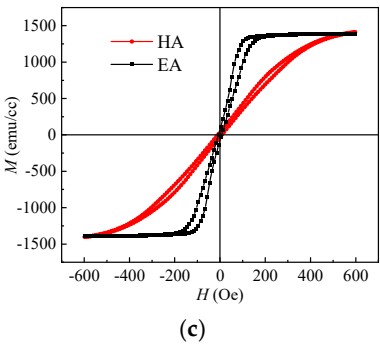

(**a**)　　　　　　　　(**b**)　　　　　　　　(**c**)

**Figure 1.** (**a**) Schematic diagram of oblique sputtering; (**b**) cross-section SEM image of a CoFeB thin film with a thickness of 450 nm on a PI substrate ($\theta$ = 60°); (**c**) normalized hysteresis loops of CoFeB films.

The cross section of CoFeB film on PI substrate was characterized by SEM and its structure is shown in Figure 1b. The result shows that CoFeB amorphous magnetic films do not have a long-range ordered crystal structure, but they can form columnar structures. CoFeB films grown on flexible PI surfaces by oblique sputtering form a tilted columnar structure. The shape anisotropy generated by this columnar structure drives the CoFeB films to exhibit perpendicular magnetic anisotropy, resulting in the formation of stripe domains.

To probe the regulation of the shape anisotropy generated by the size effect on the microscopic magnetic domain structure of CoFeB films, CoFeB films with striped domains on PI substrates were subjected to micro-nano processing such as photolithography and ion beam etching to form specific patterns. As shown in Figure 2, striped structures with different longitudinal widths were prepared. The magnetic domain characterization shows that the stripe domain alignment direction is only slightly deflected due to the weaker

shape anisotropy generated by the stripe structure than the uniaxial magnetic anisotropy induced by oblique sputtering. As the line width of the stripe pattern decreases, the aspect ratio increases and the demagnetization factor of the film changes. Accordingly, the contrast of the stripe domains decreases to 1.094°, 0.679° and 0.317° for line widths of 20 μm, 5 μm and 2 μm, respectively.

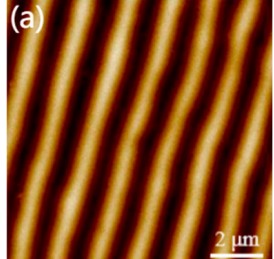 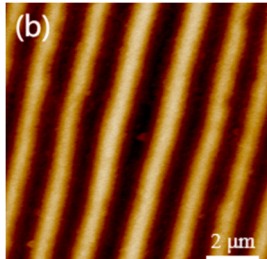 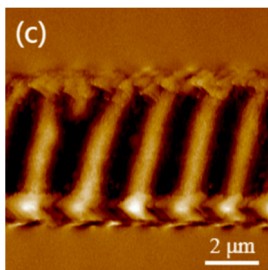 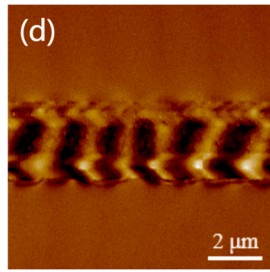 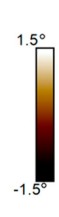

**Figure 2.** CoFeB stripe magnetic domain structure of stripe patterns with different line widths: (**a**) 100 μm × 100 μm, (**b**) 100 μm × 20 μm, (**c**) 100 μm × 5 μm, (**d**) 100 μm × 2 μm.

The regulation of the CoFeB stripe domain structure by triangular structures with different top angles $\varphi$ was investigated, as shown in Figure 3. When the top angle is large (90°) (Figure 3a), the triangular pattern does not produce any shape anisotropy along a specific direction, and there is basically no deflection in the stripe domain arrangement. At $\varphi = 60°$ (Figure 3b), there is a significant deflection of the stripe domains at the edges of the triangle, with a deflection angle of approximately 47°. At $\varphi = 30°$ (Figure 3c), the deflection angle is at its maximum, at approximately 60°. This is mainly due to the increasing anisotropy of the shape of the triangle as the angle of the top angle becomes smaller, which is aligned in the direction of the top angle and leads to a bending of the stripe domains. This effect increases progressively with decreasing size, so that the bending of the stripe domains is most pronounced at the apex of the triangle. In addition, the proposed explanation for why deflection occurs only at the edges is that the stripe domains have a large stray field at the edges and the domain structure is bent in order to minimize the overall surface demagnetization energy.

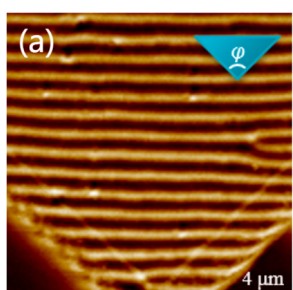 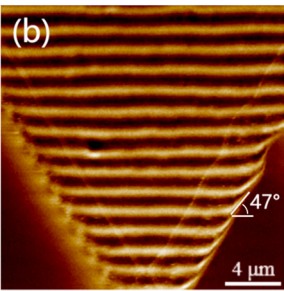 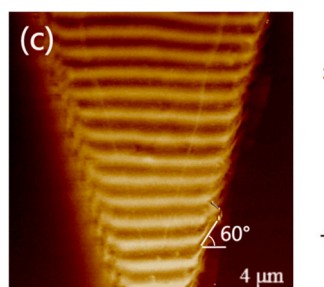 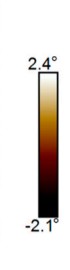

**Figure 3.** Stripe magnetic domain structure of triangular pattern CoFeB with different top angles $\varphi$: (**a**) 90°, (**b**) 60°, (**c**) 30°.

In addition, in order to design ordered magnetic domain structures with a certain local curvature, we prepared a series of rings as shown in Figure 4, all with an outer radius of 20 μm and with different ring widths. When the ring is wide (8 μm) as shown in Figure 4a, the resulting shape anisotropy is not sufficient to affect the arrangement of the striped domains. As the width decreases, the orientation of the striped domains gradually deflects. When the width decreases to 2 μm (Figure 4c), the deflection angle of the striped domains becomes larger. The local striped domains become split and show a branching structure with the overall tendency to bypass the region of lower magnetic permeability in the middle and distribute along the ring circumference.

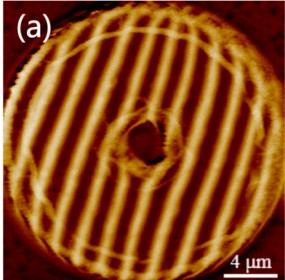 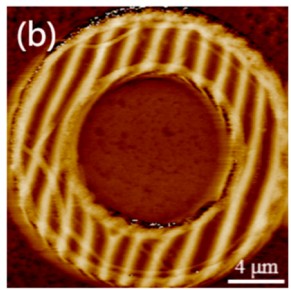 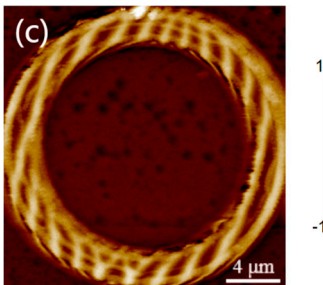

**Figure 4.** Stripe magnetic domain structure of ring-patterned CoFeB with different line widths: (**a**) 8 μm, (**b**) 4 μm, (**c**) 2 μm.

Moreover, a series of circular structures of different diameters similar to the ring were designed, as shown in Figure 5. Consistent with the pattern described previously, the circular structures at large sizes have almost no effect on the stripe magnetic domains (Figure 5a). When the size drops below a certain level (~10 μm), the circular structures only have a deflecting effect on the magnetic domain arrangement (Figure 5b). When the diameter drops to 5 μm (Figure 5c), stripe domains are deflected with large curvature and distributed along the circumference at the edges. This suggests that the size of the CoFeB microstructure needs to be close to the period of the stripe domains (~2 μm) for the size effect to be significant.

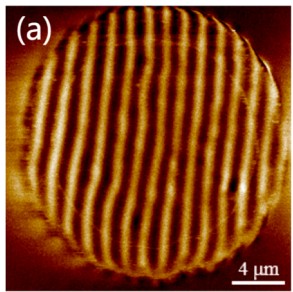 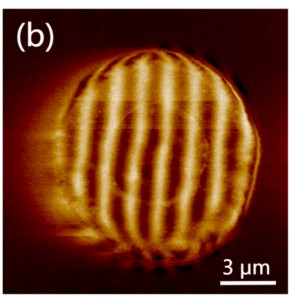 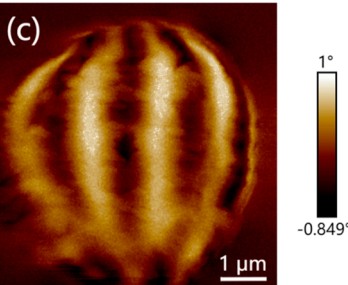 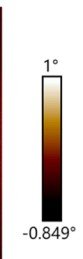

**Figure 5.** Stripe magnetic domain structure of CoFeB with circular patterns of different diameters: (**a**) 20 μm, (**b**) 10 μm, (**c**) 5 μm.

To investigate the regulation of strain on the microscopic magnetic domains of CoFeB films, we laminated the lithographed PI samples to the concave or convex curved surfaces of aluminium alloy moulds, and the radius of curvature of each mould was different, resulting in different compressive or tensile strains in the PI substrate and the CoFeB amorphous films. The strain $\varepsilon$ in the CoFeB films can be obtained from an equation:

$$\varepsilon = 1 - \frac{2r}{2r + h + t} \tag{1}$$

where $r$ is the radius of curvature of the mould (10 and 25 mm), $h$ is the thickness of the substrate (0.1 mm) and $t$ is the thickness of the CoFeB film. As shown in Figure 6, loading the substrate onto the curved surfaces with different radiuses of curvature in our work produced strains of −0.20%, 0%, 0.20% and 0.50% respectively. Figure 6a shows that the application of compressive strain to the CoFeB micro-nano structure causes the stripe domains to rotate away from the strain direction. The in-plane magnetization component decreased because of the magnetoelastic coupling effect, leading to an increase in vertical anisotropy and a significant increase in the contrast between brightness and darkness of the stripe magnetic domains, which indicates an increase in the out-of-plane stray field strength of the film. On the contrary, in the stretched state (Figure 6c,d) the striped domains tend to align parallel to the direction of tensile strain. When the strain is further increased to 0.50%, the striped domains disappear completely, which indicates that the out-of-plane

magnetization component of the striped domains gradually turns in-plane due to the tensile strain. When the domain contrast is basically zero, the magnetization component of the CoFeB film is parallel to the film surface.

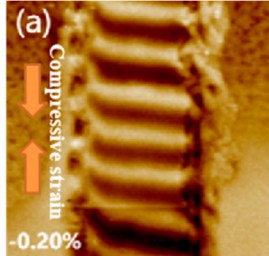 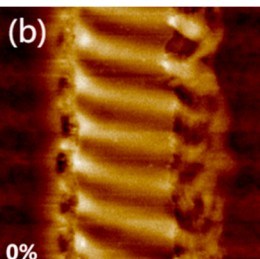 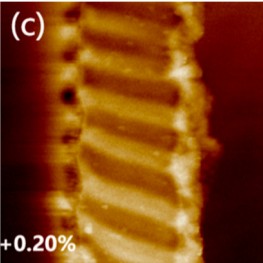 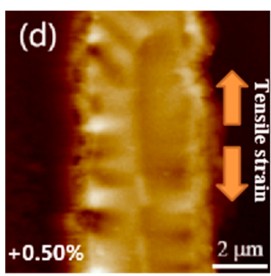 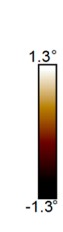

**Figure 6.** MFM images of micro-domain structure of 5 μm linewidth CoFeB thin films under different strain conditions: (**a**) 0.20% compressive strain, (**b**) no strain, (**c**) 0.20% tensile strain, (**d**) 0.50% tensile strain.

## 4. Conclusions

In summary, CoFeB films with striped domains was successfully produced on flexible PI substrates by oblique sputtering. The oblique sputtering induces a columnar crystal structure in the CoFeB films, resulting in in-plane uniaxial magnetic anisotropy and increased perpendicular magnetic anisotropy. CoFeB films with specific patterns were obtained by micro-nano processing. By characterization, it was found that the lithographically prepared stripe structure reduced the stripe domain contrast. Triangular, ring and circular patterns deflected the stripe domain alignment to various degrees. The experimental results show that the deflection of the stripe domains is caused by the shape anisotropy generated by the different patterns and that the size effect is significant only when the microstructure size is close to the period of the stripe domains (~2 μm). The stripe domains have a large stray field at the edges and the bending of the domain structure generally occurs at the edges of the pattern in order to minimize the overall surface degradation energy. Due to the coupling of the strain-induced in-plane anisotropy and the inherent perpendicular magnetic anisotropy, the stripe domains deviate from the strain direction when compressive strain is applied to the CoFeB micro-nano structure, and the striped domain lining increases. In the tensile state, the striped domains tend to align parallel to the tensile strain direction, and the out-of-plane magnetization component of the striped domains rotates in-plane.

**Author Contributions:** Conceptualization, Y.X. and H.L.; Methodology, H.L., Y.X., H.Y. and H.H.; Investigation, H.L., Y.X., H.Y. and M.L.; Writing—Original Draft Preparation, H.L.; Writing—Review and Editing, Y.X. and H.L.; Supervision, Y.X.; Funding Acquisition, Y.X. and R.-W.L. All authors have read and agreed to the published version of the manuscript.

**Funding:** We acknowledge the financial support from the National Natural Science Foundation of China (No. 51931011, 52127803, 51971233, U22A20248, 92064011, 62174164), the K. C. Wong Education Foundation (No. GJTD-2020-11), the External Cooperation Program of Chinese Academy of Sciences (174433KYSB20200013), "Pioneer" and "Leading Goose" R&D Program of Zhejiang (2022C01032), and the Ningbo "2025 S&T Megaprojects" (No. 2022Z094).

**Data Availability Statement:** All data generated or analyzed during this study are included in this article.

**Conflicts of Interest:** The authors declare no conflict of interest.

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
