# Peer review of "The Effect of Size and Strain on Micro Stripe Magnetic Domain Structure of CoFeB Thin Films"

_metals, doi:10.3390/met13040678_

Round 1

Reviewer 1 Report

In this article, the authors have carried out a very comprehensive study of the magnetic response of a CoFeB ferromagnetic film in polyimide, obtained by oblique sputtering. In it, the conclusions are supported by the results, which are clearly shown.

Author Response

Response: Thanks for your careful reading of the manuscript and positive comments on the contribution of this work to the research field.

Reviewer 2 Report

The current work illustrate the effect of size and strain in micro stripe magnetic domain structure of CoFeB thin films. I found   the work is very interesting and will add a good contribution to the scientific community. Thus, I recommend to accept the manuscript to be published in Metals, after the authors maintained these necessary points;

The introduction:

The introduction needs to be enhanced with relevant references which have an improving the perpendicular magnetic anisotropy in pure ferromagnetic materials and their alloys. The authors can use these suggested references to enhance the introduction;

1-    Salaheldeen, M.; Mendez, M.; Vega, V.; Fernández, A.; Prida, V.M. Tuning Nanohole Sizes in Ni Hexagonal Antidot Arrays: Large Perpendicular Magnetic Anisotropy for Spintronic Applications. ACS Appl. Nano Mater. 20192, 1866–1875.

2-    Salaheldeen, M.; Goyeneche, M.L.M.; Alvarez-Alonso, P.; Fernandez, A. Enhancement the perpendicular magnetic anisotropy of nanopatterned hard/soft bilayer magnetic antidot arrays for spintronic application. Nanotechnology 202031, 485708.

3-    Silva, A.S., Sá, S.P., Bunyaev, S.A. et al. Dynamical behaviour of ultrathin [CoFeB (tCoFeB)/Pd] films with perpendicular magnetic anisotropy. Sci Rep 11, 43 (2021). https://doi.org/10.1038/s41598-020-79632-0.

4-    Salaheldeen, M.; Nafady, A.; Abu-Dief, A.M.; Díaz Crespo, R.; Fernández-García, M.P.; Andrés, J.P.; López Antón, R.; Blanco, J.A.; Álvarez-Alonso, P. Enhancement of Exchange Bias and Perpendicular Magnetic Anisotropy in CoO/Co Multilayer Thin Films by Tuning the Alumina Template Nanohole Size. Nanomaterials 202212, 2544. https://doi.org/10.3390/nano12152544.

5-      https://doi.org/10.1063/5.0106414

Materials and methos:

The authors should add more information and details in the experimental part and how they estimate the layer thickness of the film. If possible, could add the commercials sources of the materials and targets are used in the current work. In addition, should the author describe more details about the sputtering condition and if they applied any magnetic field during the deposition process.

Results:

The authors indicated several times in the manuscript that CoFeB has amorphous structure, If possible, could please add the XRD analysis to proof that, as there are many research indicated that the as deposited CoFeB van has a polycrystalline nature for samples deposited directly on the glass substrate at room temperature.

For more accurate values could the authors plot the M—H loops with term od emu/CC instead of normalized loops.

The way to check the magnetic anisotropy is unclear, -I do not understand they speak about the in-plane anisotropy, so, if possible, angular M-H loops should be done. To check the in-plane magnetic anisotropy and determined the uniaxial in-plane magnetic anisotropy.

The author should declare the way to obtain the hard and easy axis. The magnetic anisotropy is expected to be in-plane.

Reviewer 3 Report

The article is devoted to the study of the technology of manufacturing flexible magnetic films and the evolution of film properties during deformation, as well as to the study of various properties and the dependence of their changes depending on external influences. The article has a fairly large scientific potential and practical significance, since the proposed technology for producing flexible magnetic films using sputtering. In general, the presented data correspond to the subject of the declared journal and can make a significant contribution to the development of this area of research. The article can be accepted for publication after the authors answer the reviewer's questions, as well as provide detailed answers and highlight them in the text.

1. The authors should provide a more detailed description of the relevance and novelty of the presented study, as well as provide comparative data with other types of thin films.

2. The authors should explain what exactly is the choice of polyimide films for obtaining amorphous magnetic films? Is this related to the further practical application of the obtained magnetic films in view of the stability of polyimide?

3. The authors should give an explanation of how exactly the amorphization of films and their magnetic anisotropy of films are related when the synthesis conditions are varied?

4. The presented results of morphological features require additional morphological studies and a detailed presentation of the data obtained using the results of high-resolution scanning electron microscopy.

5. The authors should confirm the amorphism of the obtained thin films using X-ray diffraction analysis methods.

Round 2

Reviewer 2 Report

The authors fixed all the comments and I recommend to accept the manuscript at the present form.